# Goals of older hospitalised patients: a qualitative descriptive study

Maria Johanna van der Kluit,[1] Geke J Dijkstra,[2] Sophia E de Rooij[1]

**To cite:** van der Kluit MJ, Dijkstra GJ, de Rooij SE. Goals of older hospitalised patients: a qualitative descriptive study. *BMJ Open* 2019;**9**:e029993. doi:10.1136/bmjopen-2019-029993

¹University of Groningen, University Medical Center Groningen, University Center for Geriatric Medicine, Groningen, The Netherlands
²University of Groningen, University Medical Center Groningen, Department of Health Sciences, Applied Health Research, Groningen, The Netherlands

**Correspondence to**
Mrs Maria Johanna van der Kluit; m.j.van.der.kluit@umcg.nl

## ABSTRACT

**Objectives** Since the population continues ageing and the number of patients with multiple chronic diseases is rising in Western countries, a shift is recommended from disease oriented towards goal-oriented healthcare. As little is known about individual goals and preferences of older hospitalised patients, the aim of this study is to elucidate the goals of a diverse group of older hospitalised patients.

**Design** Qualitative descriptive method with open interviews analysed with inductive content analysis.

**Setting** A university teaching hospital and a regional teaching hospital.

**Participants** Twenty-eight hospitalised patients aged 70 years and older.

**Results** Some older hospitalised patients initially had difficulties describing concrete goals, but after probing all were able to state more concrete goals. A great diversity of goals were categorised into wanting to know what the matter is, controlling disease, staying alive, improving condition, alleviating complaints, improving daily functioning, improving/maintaining social functioning, resuming work/hobbies and regaining/maintaining autonomy.

**Conclusions** Older hospitalised patients have a diversity of goals in different domains. Discussing goals with older patients is not a common practice yet. Timely discussions about goals should be encouraged because individual goals are not self-evident and this discussion can guide decision making, especially in patients with multimorbidity and frailty. Aids can be helpful to facilitate the discussion about goals and evaluate the outcomes of hospitalisation.

## BACKGROUND

Since the population continues ageing and the number of patients with chronic diseases is rising in Western countries, a shift is recommended from disease oriented towards goal-oriented healthcare. Questioned is whether healthcare always aims for the desired outcomes for patients.[1–3]

Goals are the personal health and life outcomes that people hope to achieve through their healthcare.[3] Little is known about the individual goals and preferences of older hospitalised patients. Observations by a phenomenological researcher revealed that the main concerns for older hospitalised patients were whether they would be able again to carry out activities that were important to them such as working on the allotment, attending the wedding of a

### Strengths and limitations of this study

► Qualitative descriptive research stays close to the perspective of the older patient.
► We interviewed a broad variety of older patients during their hospitalisation, in a real life situation.
► It is difficult to reach saturation on level of goals. Although the categories became clear, there might always emerge new specific individual goals when approaching new patients.

granddaughter or whether they would be able to live at home again. Older patients, however, seldom spoke spontaneously about this with their care professionals.[4]

The need for and emphasis on social and physical activities and to live at home are also reflected in other studies. A study into patient goals after aortic aneurysm repair revealed that patients prioritise functional outcomes and recovery time after the operation, as well as energy levels, pain and the ability to walk again. In this study, recovery time was found more important than survival.[5] This was also seen in a study into patient goals of the treatment of severe aortic stenosis. In that study, patients prioritised to be able to perform activities again such as hobbies or social activities, followed by remaining independent. Staying alive had the lowest priority for most patients.[6] Since older hospitalised patients form a heterogeneous group because of the reason for hospitalisation, comorbidities, polypharmacy, disabilities and social background, the aim of this study is to elucidate the goals of a broad group of older patients hospitalised for medical or surgical reasons.

## METHODS

To take account of the perspective of the older patients, a qualitative descriptive method was used.[7 8]

### Population

Patients were recruited during their hospitalisation in a university teaching hospital in the

northern part of the Netherlands and a regional teaching hospital in the central part of the Netherlands.

Inclusion criteria were as follows: (1) hospitalisation expected for at least 48 hours; (2) aged 70 years and older; (3) being able to speak and understand Dutch; (4) not expected to die within the next 48 hours; (5) informed consent to the interview and audio-recording.

A purposive sample was used. Within the group of eligible patients, we aimed for maximum variation in age, frailty, living at home or in a nursing home, planned and unplanned admissions, university hospital or regional hospital. Frailty was determined by the Fried criteria as operationalised by Avila-Funes et al[9] and asked to the patient himself.

Data gathering and analysis were alternated. We aimed to continue sampling until saturation was achieved, meaning no new information emerged from the patients. Since it appeared during the study difficult to reach saturation on goal level, we decided to aim for saturation on category level.

In total, 28 patients were interviewed. Details of the sample are shown in table 1.

### Data collection

After establishing inclusion criteria by the staff nurse, eligible patients were given an information letter and were approached by the interviewer (MJvdK) for further information about the procedure and to obtain informed consent during their hospitalisation. The Medical Ethics Research Committee of the University Medical Center Groningen confirmed that the Medical Research Involving Human Subjects Act did not apply to the research project. Official approval by the committee was hence not required.

Open interviews were conducted during hospitalisation by MJvdK. MJvdK is an experienced nurse, but not working as a nurse in the hospitals where the interviews took place. MJvdK is trained in qualitative research and interviewing. To comfort the patient, the interviews started with giving the patient the opportunity to explain the reason for hospitalisation. After that, the main question posed by the interviewer was: What do you hope to accomplish with this hospitalisation? Probes were used to clarify the goals of the participants, like 'what do you mean with…', 'can you give an example of…', summarising. The interviews took place in the patient's room or, when the patient shared a room, in a family or examination room on the ward. The interviews took 15 to 60 min and were audio-recorded and transcribed verbatim. After each interview, an interview memo was written to gather initial impressions of the interview.

### Analysis

Since little is known about the goals of older hospitalised patients, an inductive content analysis was used.[10 11]

Data gathering and data analysis were alternated. The analysis started with open coding; the codes were then grouped

| Table 1 | Patient characteristics |
|---|---|
| **Gender** | |
| Male | 16 |
| Female | 12 |
| **Age (years)** | |
| 70–79 | 14 |
| 80–89 | 11 |
| 90–99 | 3 |
| **Frailty** | |
| Non-frail | 11 |
| Frail | 17 |
| **Living situation** | |
| At home | 22 |
| Senior home | 3 |
| Nursing home | 3 |
| **Hospital** | |
| University | 26 |
| Regional | 2 |
| **Admission day interview** | |
| <3 days | 5 |
| 3–5 days | 16 |
| 6–10 days | 4 |
| >10 days | 3 |
| **Specialism** | |
| Internal medicine | 20 |
| Surgery | 5 |
| Cardiology | 3 |
| **Admission due to*** | |
| Dyspnoea | 7 |
| Constipation | 3 |
| Malignancy | 3 |
| Fall | 2 |
| Swollen leg | 2 |
| General malaise | 2 |
| Abdominal pain | 2 |
| Diarrhoea | 2 |
| Vomiting | 1 |
| Infection device | 1 |
| Myocardial infarction | 1 |
| Aorta surgery | 1 |
| Transcatheter aortic valve replacement | 1 |
| **Type of admission** | |
| Acute | 23 |
| Planned | 5 |

*Admission reason according to patient interview.

into categories and data were compared within and between categories and the categories were described.[10]

All transcripts were read by the first (MJvdK) and second (GJD) authors independently and then the goals and codes were compared. The grouping of the codes into categories was also done by the first and second authors independently, the differences were then discussed and solved by consensus.

During the entire process, memos were written about the interviews and coding process. Data analysis and organisation were supported by the use of Atlast.ti V.5.2.18.

Interviews and analysis were all in Dutch. The categories, codes and quotes were translated into English in the final stage and checked and edited by a native English speaker.

### Patient and public involvement

Patients or public were not involved in the design and conduct of this study.

### RESULTS

After the question 'What do you hope to accomplish with this hospitalisation?', some participants replied with clear, concrete answers while others initially started with broad, abstract answers like 'getting better' and 'recovering'. With probing, all participants were able to explain what, for example, 'getting better' meant for them and were able to state more concrete goals, except for one patient with delirium.

For example:

Interviewer: Because what is your goal with this hospitalisation? Patient: Goal? Interviewer: Yes. Patient: That I am getting better. Interviewer: And what is better for you, can you describe that? Patient: Yes, that I … well … get my appetite back and drink well, because I am not interested in whether I get anything or not at the moment. I am not hungry, I am not thirsty and that has to change. Interviewer: Yes. Patient: And if I then grow stronger again. I have lost a lot of weight. From 88 to 82, I believe. Interviewer: In how much time? Patient: About a week. I was still very weak yesterday. Interviewer: Yes Yes. So grow stronger. Patient: To grow stronger. And that I am back on my feet, that I can walk with a crutch and I'm done here as soon as possible and that I can go back home. That is my goal. (P11, 89 years, acute admission, internal medicine, frail)

The goals patients had were grouped into the following categories: wanting to know what the matter is, controlling disease, staying alive, improving condition, alleviating complaints, improving daily functioning, improving/maintaining social functioning, resuming work/hobbies, regaining/maintaining autonomy (table 2). In table 3, preferences patterns and examples per patient are shown.

| Table 2 | List of categories and codes |
| --- | --- |
| **Categories** | **Codes** |
| Wanting to know what the matter is | ▶ Finding cause of complaints<br>▶ Ruling out severe affairs |
| Controlling disease | ▶ Curing<br>▶ Slowing down progression of the disease |
| Staying alive | ▶ Staying alive |
| Improving condition | ▶ Improving condition<br>▶ Increasing energy<br>▶ Feeling better<br>▶ Reducing uncertainty<br>▶ Regaining weight |
| Alleviating complaints | ▶ Reducing/eliminating pain<br>▶ Reducing shortness of breath<br>▶ Stopping vomiting<br>▶ Reducing dizziness<br>▶ Restoring stools<br>▶ Reducing sweating<br>▶ Restoring appetite<br>▶ Restoring sleep |
| Improving daily functioning | ▶ General functioning<br>▶ Walking<br>▶ Moving<br>▶ Housekeeping<br>▶ Shopping<br>▶ Cooking<br>▶ Self-care |
| Improving/maintaining social functioning | ▶ Visiting family/friends<br>▶ Making a day trip<br>▶ Enjoying the presence of partner/children |
| Resuming work/hobbies | ▶ Resuming (volunteer) work<br>▶ Gardening<br>▶ Resuming hobbies<br>▶ Resuming sport |
| Regaining/maintaining autonomy | ▶ Going back home<br>▶ (Re)gaining freedom<br>▶ Regaining/ maintaining independence |

### Wanting to know what the matter is

Several patients indicated that they wanted to know what was the cause for their complaints or the patient wanted to rule out severe other explanations. For example:

That pain is caused by something. And I would really like to know what that is. (P22, 74 years, acute admission, internal medicine, non-frail)

### Controlling disease

The group 'Controlling disease' is used for medical control of diseases. Some patients aimed for complete cure, like people with cancer. But for most the goal was to stop or slow down the disease progression because they knew their chronic condition was not curable. For example:

**Table 3** Preference patterns and examples per participant

| Participant | Matter | Controlling disease | Staying alive | Improving condition | Alleviating complaints | Daily functioning | Social functioning | Work/hobbies | Autonomy | Example quote |
|---|---|---|---|---|---|---|---|---|---|---|
| 01 | X | | | | | | | X | | P: My objective is, actually, of course that I uh, recover completely from those uh, defects that I am currently experiencing. That I could do the things again that I do now every day. I: Yes. And what are they? P: And those are many things. On Sunday morning I walk with a couple of women. Then I walk through the heath and then I walk for an hour and then uh. To maintain my condition. And I always maintain that condition. I'm always busy with that kind of nonsense. Nonsense, well, yes. It limits what I want. Yes, so I think uh, I like doing that. Hey? Just as much as that I like to play tennis. And stand in front of the net and can give a ball a swipe the moment it comes up to me and then place it neatly. Well those are all things. They all play a role. |
| 02 | | | X | | | X | X | | | No, I had to stay alive. I felt. And nothing more. I mean, yes, no, that is, of course, everything. |
| 03 | | | | X | | X | | | X | Well, walking, moving, covering more distance and more. A better condition. |
| 04 | | | | | X | X | | X | | P: Yes, that I could function normally again. Yes. I: And what then are the things that are important for your functioning? P: Well, that I can just do my homework again. I don't have to do anything else. Work a little in the garden, things like that. That. I think that is important, definitely. |
| 05 | | | | | | | X | X | X | Well, in my own house, of course! |
| 06 | | | | X | X | | | | X | P: That it becomes a little easier. I: And what should become easier? P: That shortness of breath. |
| 07 | X | | | | | X | | | X | I, I wanted to know what the matter was. And that, uh, they couldn't judge that from here. |
| 08 | X | | | X | X | | X | X | X | Well, that I am getting fit again and have no pain. And that no other annoying things come to light. |
| 09 | X | | | | X | | X | X | X | That I am going to get a bit more of my, my freedom. Yes, there is nothing worse if you can't go to the toilet. |
| 10 | | | | X | X | X | | | X | Yes, that I … well … get my appetite back and drink well, because I am not interested in whether I get anything or not at the moment. I am not hungry. I am not thirsty and that has to change. And if I then grow stronger again. I have lost a lot of weight. |
| 11 | | | | | | X | | | | Well, that in any case, that I, uh again, will be a little more agile and so hey. Yes. |
| 12 | | | | | X | X | | | X | I would like to keep what independence I had. |
| 13 | | X | | | | X | | | | That the process of … Or the consequences of the diabetes, that those will be stopped, eh. That it does not get worse or that the sugars are all the time too high. |

Continued

**Table 3** Continued

| Participant | Matter | Controlling disease | Staying alive | Improving condition | Alleviating complaints | Daily functioning | Social functioning | Work/ hobbies | Autonomy | Example quote |
|---|---|---|---|---|---|---|---|---|---|---|
| 14 | | X | X | | X | | | | X | P: Well that it will be a little bit better and I can go along a bit. I: That you can go along a bit? What do you mean by that? P: Yes well, that I am alive, so to speak. |
| 15 | | | X | X | | | | | | P: Getting better and... I: Getting better you say. And what is "better" for you? P: That, I, say, could compete again. |
| 16 | | | | | | | X | X | X | Well, that I can just, uh, just be home again. And I, uh, still play cards always, and I really like that. |
| 17 | X | | | | X | | X | X | X | That diarrhoea must stop. That's what it's all about. |
| 18 | | | | | X | X | | | | Well, that I get rid of that shortness of breath. |
| 19 | | X | | | | | | | | That I, that little bit kidney that I have, that I can keep that. That's what I hope to achieve. |
| 20 | | | X | | | | | X | | Well, still live tomorrow and the day after tomorrow. So, uh, I am, what's that called, from 1922 and because of this pacemaker, I don't know if, but my expectations might be too high. But I'm going to live for a few more years because of this pacemaker. |
| 21 | | X | | | | | X | X | | Sitting at my desk and writing. Once in a while, when my wife is driving the car, going out for dinner or having a drink somewhere. Family visits. |
| 22 | X | | | | X | | | | | That pain is caused by something. And I would really like to know what that is. |
| 23 | | | | | X | X | | X | | Just without pain, uh, not vomiting. Function normally. Uh, I'm 70, but I'm still active. I am a forester and, uh, I coordinate the volunteers on the estate. |
| 24 | | X | X | | | | X | X | | Simply, cosy and nice, living on. And we had it very good, yes, with our family. |
| 25 | | X | X | | | X | | | X | That my, that that bacterium is being fought enough to be able to live on again, or at least that it is gone and that I can just go back to my house and work again. |
| 26 | | X | | | X | X | X | | X | The main goal for me is that the pain goes away and that I largely stop using those medications. |
| 27 | | | | X | X | X | X | X | | Well, to go out for a change and enjoy yourself. And visiting friends again. They visited us, but you also want to go out yourself for a change. And I didn't do that anymore at all. |
| 28 | | X | | | X | X | X | X | X | And the aim is then simply to get that again, yes, so that you can walk well on that foot again. Yes and that you can make all movements pretty much, right? And not getting extra wear, which only makes it worse. |

That the process of … Or the consequences of the diabetes, that those will be stopped, eh. That it does not get worse or that the sugars are all the time too high. (P13, 71 years, planned admission, surgery, frail)

### Staying alive

Several patients stated that they hoped to stay alive or to live a few more years due to hospital admission. For some patients, the argument to stay alive was the main reason to go to hospital. For example:

No, I had to stay alive. I felt. And nothing more. I mean, yes, no, that is, of course, everything. (P2, 88 years, acute admission, internal medicine, non-frail)

### Improving condition

This category is a subjective experience by the patient and contains codes like improving condition, augmenting energy, feeling better, reducing uncertainty and regaining weight. For example:

Patient: Yes, enhancing condition and that I can cope a bit more, actually much more. But yes, that I have to, to, to play a football match, no, that time does not return. Interviewer: That is pretty far-fetched? And what would be a realistic goal for you? Patient: Being able to walk a bit more decently, and sustaining, my fitness, building that up again. Yes, to be able to do a little bit more conditionally. (P3, 70 years, acute admission, internal medicine, frail)

### Alleviating complaints

A broad variety of complaints were described, which participants wished to alleviate, including pain, shortness of breath, vomiting, dizziness, obstipation, diarrhoea, sweating, lack of appetite and insomnia. For example:

That diarrhoea must stop. That's what it's all about. (P17, 88 years, acute admission, internal medicine, frail)

### Improving daily functioning

While some patients stated improving functioning in general, others named specific functions like walking, moving, housekeeping, shopping, cooking and self-care. For example:

That I can function independently again with a walker. (P7, 82 years, acute admission, internal medicine, frail)

### Improving/maintaining social functioning

Participants mentioned various social activities they wanted to be able to participate in again, like visiting family or friends or making a day trip. For example:

Meeting friends and taking a drive around and perhaps drink a cup of tea somewhere, it does not have to be luxurious or fancy at all. But enjoying things.

Going to the theatre once and yes, those things. (P8, 86 years, acute admission, internal medicine, frail)

For some, just enjoying the presence of their partner and close family members was very important.

### Resuming work/hobbies

Several participants indicated that they wanted to resume their work, for example, volunteer work, assisting in the family business or scientific work. Others wanted to resume their sports, working in the garden or hobbies. For example:

And, uh, now I hope to achieve, that I can go outside more and enjoy my garden too, because I love gardening a lot and so, that was all gone. (P27, 72 years, planned admission, cardiology, frail)

### Regaining or maintaining autonomy

This category was used for statements of participants about maintaining or regaining their independence or freedom. Also the code 'going back to own house', was placed into this category. For example:

Yes, a bit more freedom, going somewhere alone once again. Yes, I just can't. (…) Yes, then I have to take a taxi. Yes, then I also lost my freedom. Because then you also need certain … And I love my freedom. If I want to go somewhere, I have to be able to do that. And not arranging everything in advance. (P26, 74 years, planned admission, surgery, frail)

### DISCUSSION

As far as we know, this is the first study investigating the goals of already hospitalised older patients admitted for a broad diversity of reasons. It was remarkable that some patients initially had difficulties stating concrete goals, but after probing all were able to state more concrete goals.

Patients reported a variety of goals, which could be grouped into the categories 'wanting to know what the matter is', 'controlling disease', 'staying alive', 'improving condition', 'alleviating complaints', 'improving daily functioning', 'improving/maintaining social functioning', 'resuming work/hobbies' and 'regaining/maintaining autonomy'.

Since we used an inductive method, our categorisation is different from other studies, but also showed some similarities. Coylewright *et al* categorised the goals of older adults eligible for an aortic valve replacement into the groups: 'staying alive', 'reducing/eliminating pain or symptoms', 'maintaining independence' and 'ability to do a specific activity'.[6] This categorisation has many similarities with the categories we constructed, although ours were more detailed.

Goals of community-dwelling older adults were placed in the categories 'health problems', 'mobility', 'emotions', 'independence and autonomy', 'social and

family relationships', 'activities', 'living accommodation', 'healthcare services' and 'finances'.[12]

Vermunt *et al* investigated patient goals from the perspective of general practitioners and geriatricians and came to the following categories: 'fundamental goals', 'functional goals' and 'disease-specific or symptom-specific goals'.[13] Again our categorisation has similarities, but is more detailed.

The goals set during hospitalisation also are in line with what community-dwelling older adults find important in quality of life or well-being, namely 'staying independent', 'social life', 'hobbies', 'activities', 'health' and 'own environment'.[14 15] Apparently, hospitalisation is seen by patients as an option to improve or maintain quality of life or well-being.

Setting goals is not yet common practice, not from the perspective of the patient, nor from the healthcare professional. This could be explained because historically patients presented with acute problems and it was expected that the healthcare professional would solve the acute problem and the patient would return to a normal healthy state. However, nowadays many complaints of older patients are caused by, often multiple, chronic diseases, which can only be controlled but not completely cured. Probably this shift still has not entered completely into daily clinical practice.[16] Several other barriers for discussing goals are described, including considering talking about personal goals impertinent, lack of skills by healthcare professionals, focus on symptoms, limited time and the presumption by both patients and healthcare professionals that all patients have the same goals.[16] There are, however, several examples that rebut this last presumption.[13 17 18] Therefore, it is important to discuss individual goals explicitly with the patient, which can also guide decision making in case of multimorbidity and provide important information for handling acute health situations in future.[13]

### Strengths and limitations

The strengths of our study include that we interviewed older patients during their hospitalisation, in a real life situation, at different hospital wards, and we included a broad variety of patients. This led to a broad overview of categories of goals, but did not lead to very specific individual goals. Another limitation is that it is difficult to reach saturation on level of goals. Although the categories became clear, there might always emerge new specific individual goals when approaching new patients.

### Conclusion

Older hospitalised patients have a diversity of goals in different domains. Discussing goals with older hospital patients is not common practice yet, and many patients and healthcare professionals are not familiar with discussing personal goals. Timely discussions about goals should be encouraged because individual goals are not self-evident and this discussion can guide decision making, especially in patients with multimorbidity and frailty. Aids are needed to facilitate the discussion about goals and the evaluation of goals of hospitalisation.

**Acknowledgements** We would like to thank all interviewees for participating in this study and sharing their stories with us and Daniël Bosold for his help with text editing.

**Contributors** MJvdK designed the study and conducted the interviews. MJvdK and GJD read all transcriptions. MJvdK predominantly performed the qualitative analysis. As part of this analysis, MJvdK and GJD regularly discussed the coding process and categorisation. MJvdK wrote the first draft of the manuscript; GJD and SEdR contributed significantly to subsequent manuscript revisions. All authors read and approved the final version of the manuscript.

**Funding** This study was funded by an unrestricted grant from the University of Groningen.

**Competing interests** None declared.

**Patient consent for publication** Not required.

**Provenance and peer review** Not commissioned; externally peer reviewed.

**Data availability statement** The results in this paper are based on the transcripts of the recorded audio interviews with patients. Data supporting the findings of this study are found in the translated quotes as seen in the Results section of this article. However, to protect the participants' identities, the full data of this study (transcripts and audio files) will not be made available to the public.

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
