## [Reviewer comments · BMJ Open]

ARTICLE DETAILS

TITLE (PROVISIONAL)	Goals of older hospitalised patients: A qualitative descriptive study.
AUTHORS	van der Kluit, Maria; Dijkstra, Geke; De Rooij, Sophia

VERSION 1 - REVIEW

REVIEWER	Mary Tinetti Yale University School of Medicine, USA
REVIEW RETURNED	04-Mar-2019

GENERAL COMMENTS	Identifying patients health goals and incorporating them into decision making is an important and relevant topic. INTRODUCTION In the second paragraph, authors note that, " Observations revealed that the main concerns for older hospitalised patients were whether they would be able again to carry out activities that were important to them such as working on the allotment, attending the wedding of a granddaughter or whether they would be able to live at home again." It is not clear whose observations these are. Please clarify this statement. METHODS In the Introduction, authors note, "Since older hospitalised patients form a heterogeneous group because of the reason for hospitalisation, comorbidities, polypharmacy, disabilities and social background, the aim of this study is to elucidate the goals of a broad group of older patients hospitalised for medical or surgical reasons." Interviewing only 28 individuals seems inconsistent with this aim. If the idea was to determine the goals of a broad spectrum of patients, why not conduct enough interviews from people in each of their predetermined these groups to address their aim? For example, only 3 were from senior homes and nursing homes. Only 2 were from the nonacademic hospital. Only 5 were on surgical, and 3 on the cardiology service. It isn't clear how patients who met criteria were selected or invited. Presumably there were many more than 28 persons who met criteria. Please clarify why these 28 and from how many eligible persons were they chosen?
--

	The opening question is very broad and vague. What were examples of probes that might help focus on specific goals? RESULTS Investigators report that, "...all were able to state more concrete goals" However, many of the categories and codes remain general such as: wanting to know what the matter is; controlling disease; staying alive; improving condition; increasing quality of life. We usually think of goals and specific and actionable, e.g. "Enough relief of pain that I can walk at least a block when I get home." The alleviating symptom and functioning categories seem the most actionable based on the quotes provided. With only 28 patients, it would be manageable, interesting, and helpful to identify categories and supportive quotes for each of them. Comparing groups – The numbers of many categories were too small to make any comparisons. Such an analysis would presumably require saturation in each designated category. I would suggest eliminating this section. DISCUSSION As noted above, many of the presumed "concrete goals" were vague and more reflective of general values rather than actionable and specific goals. Authors should clarify why they feel these are concrete goals or acknowledge the lack of specificity and actionability for many of them. This point is relevant because Investigators note that goals elicited can help inform care. Please address this issue for readers.
--	--

REVIEWER	Christiane Muth Institute of General Practice, Johann Wolfgang Goethe University, Frankfurt / Main, Germany
REVIEW RETURNED	15-Mar-2019

GENERAL COMMENTS	The manuscript addresses a relevant topic of increasing awareness, as goal orientation is a necessary ingredient to patient-centred care. It is clearly and concise written, although I had some doubts (e.g., whether "freedom" was the correct translation of a verbatim or whether "autonomy" would suit better; as I am a non-native, I'd prefer to leave this open to a native). Also, the formal reporting quality was fine and in accordance with recommended standards. Major amendments: 1. Introduction section: The authors claim that their investigation was the first in hospital setting. Formally, this may be but what did the authors expect to find different/special in their setting in comparison to other settings (e.g., primary care and long-term care facilities)? In other words, what is already known about older patients' goals from other settings and what was the expected additional value to investigate patients' goals in hospital care? The background section could be improved by a more profound search for and summary of existing evidence (e.g., Vermunt et al. 2017, PMID: 28760149) to derive the rationale of the authors' objective.
---

	2. Methods: Closely related to the issue of the setting is the approach to individual study participants. How did the authors recruit patients and what was the context when they asked them for participation (e.g., a day after admission, the day before discharge or at variable situations)? This may also explain why a patient gave a written informed consent but could not be interviewed due to a delirium. Did the authors intend to include patients with planned and unplanned admission and if so, why? 3. Methods: The authors aimed at a saturated sample - what was the criterion for saturation? If the criterion was about patients' goals, was it realistic to achieve it (and a suitable sampling frame, therefore)? 4. Methods / introduction: what was the authors' definition of goals? Did they test / pilot their key question to assure that patients understand it? 5. Methods: How was frailty assessed in the sample? 6. Methods / results / discussion: what do the authors mean with "probing"? What did they do to achieve answers about (health-related) goals from patients when interviews took up to 60 minutes? As this is also an important result, why didn't they present details from this "probing"-approach? Presented results could be further discussed - did the authors face similar difficulties like Boeckxstaens et al. 2016 (PMID: 29090183) and if so, what is the impact for clinical implementation in goal setting? 7. Results: Why did the authors decide to present this large number of categories, although there are some overlaps which might encourage other researchers to further collapse them (e.g. categories on controlling disease and improving conditions, on daily functioning and resuming hobbies / work, where codings list different types of (instrumental) activities of daily life)? 8. Results: The presentation of the verbatims may support the results on 'comparing groups' better when more characteristics would be provided than just an ID and the age (e.g. whether this patient had a planned or unplanned admission). Further, I suppose that interviews were taken in Dutch but verbatims are presented in English - at what stage of analysis / reporting the verbatims were translated and by whom? 9. Discussion - strengths and limitations: What may be the impact of a potential selection bias due to the recruitment / sampling process (> 90% of patients were included from the university hospital but <10% from the other hospital) on the results? Additional comment: I am not familiar with rules and regulations in the Netherlands and was surprised that an ethics vote was not required. Therefore, I am not sure, whether any further data protection should be taken into consideration (to use pseudonyms not only for patients but also for the hospital names).
--	---

VERSION 1 – AUTHOR RESPONSE

Reviewer: 1

Reviewer Name: Mary Tinetti

Institution and Country: Yale University School of Medicine, USA Please state any competing interests or state 'None declared': None

Please leave your comments for the authors below

Identifying patients health goals and incorporating them into decision making is an important and relevant topic.

INTRODUCTION

In the second paragraph, authors note that, "Observations revealed that the main concerns for older hospitalised patients were whether they would be able again to carry out activities that were important to them such as working on the allotment, attending the wedding of a granddaughter or whether they would be able to live at home again." It is not clear whose observations these are. Please clarify this statement.

Response: We thank the reviewer for allowing us to clarify this: These were observations through shadowing by a phenomenological researcher. We added this information in the text.

METHODS

In the Introduction, authors note, "Since older hospitalised patients form a heterogeneous group because of the reason for hospitalisation, comorbidities, polypharmacy, disabilities and social background, the aim of this study is to elucidate the goals of a broad group of older patients hospitalised for medical or surgical reasons." Interviewing only 28 individuals seems inconsistent with this aim. If the idea was to determine the goals of a broad spectrum of patients, why not conduct enough interviews from people in each of their predetermined these groups to address their aim? For example, only 3 were from senior homes and nursing homes. Only 2 were from the nonacademic hospital. Only 5 were on surgical, and 3 on the cardiology service.

It isn't clear how patients who met criteria were selected or invited. Presumably there were many more than 28 persons who met criteria. Please clarify why these 28 and from how many eligible persons were they chosen?

Response: We used a purposive sampling strategy with maximum variation sampling. We have to apologize that the word "maximum" disappeared from the text in the review process. For clarity reasons we added this in the text. The goal of maximum variation sampling is "capturing and describing the central themes that cut across a great deal of variation" (Patton, 2002) Characteristic for purposeful sampling is that it is not sought for statistical representativeness, but informational representativeness. Only enough people from a certain group are needed to obtain enough information (Sandelowski, 1995). The question is: what is enough people. Our criterion was enough to reach saturation, which we did on category level.

Regarding the way selection of participants took place: Although there were many eligible admissions, since admissions are a continuous process in hospitals, we selected our participants purposefully and not randomly. We added some extra information in the population paragraph to make clear that data gathering and analysis were alternated and formed a continuous process. Analysis informed new sampling. Some patient characteristics were quite rare, like patients from nursing homes, so that we

had to wait a few weeks before we found a new participant. Patients were therefore “handpicked” based on the mentioned characteristics.

The opening question is very broad and vague. What were examples of probes that might help focus on specific goals?

Response: We added some examples, for example: “what do you mean with...”, “can you give an example of...”, summarizing, we added an extra quote with probes in the results section. Besides, the interviewer used many continuing quotes like yes, mmm-hmms, and non-verbal signs like nodding, etc.

RESULTS

Investigators report that, “...all were able to state more concrete goals” However, many of the categories and codes remain general such as: wanting to know what the matter is; controlling disease; staying alive; improving condition; increasing quality of life. We usually think of goals and specific and actionable, e.g. “Enough relief of pain that I can walk at least a block when I get home.” The alleviating symptom and functioning categories seem the most actionable based on the quotes provided.

Response: There is a tension between being specific and more general. We think “more concrete” is not similar to specific. It is somewhere in between on a scale from broad/abstract to specific. The interviewer did not always question further until we received an answer as specific as the example of the reviewer. The specific example the reviewer gave, is very useful in an individual clinical setting, but not for research purposes, where we wanted to make a description of the variety of goals patients have regarding hospitalisation. Although qualitative descriptive research is the qualitative method that stays closest to the data, there is still a form of interpretation, abstraction and categorisation. As we would have stayed on such a specific action level as the reviewer suggested, it would have been impossible to reach saturation, make connections between participants and form categories. That process needs a more abstract level.

Apart from this, we have chosen an open explorative approach regarding goals of hospitalisation and not only confine ourselves beforehand to actions. Since it appeared that patients had also non actionable goals like wanting to know what the matter is, improving condition and alleviating complaints, we think this was a justifiable choice.

We agree with the reviewer regarding the broadness of the goals “enhancing quality of life” and “enjoying life”. We went back to the data and discovered that the participants who mentioned these goals were able to specify this, so this was also covered by other goals. We therefore removed these goals.

With only 28 patients, it would be manageable, interesting, and helpful to identify categories and supportive quotes for each of them.

Response: As the reviewer suggested, we made a table with all participants in which we indicated which categories were applicable for that participant and one quote per participant.

Comparing groups – The numbers of many categories were too small to make any comparisons. Such an analysis would presumably require saturation in each designated category. I would suggest eliminating this section.

Response: As the reviewer suggested, we erased this paragraph and the information about this in the Discussion and Conclusion.

DISCUSSION

As noted above, many of the presumed “concrete goals” were vague and more reflective of general values rather than actionable and specific goals. Authors should clarify why they feel these are concrete goals or acknowledge the lack of specificity and actionability for many of them. This point is relevant because Investigators note that goals elicited can help inform care. Please address this issue for readers.

Response: We agree with the reviewer that our article did not give an overview over very specific individual goals, like needed in individual clinical situations. Our aim was to give an overview of the diversity of (categories) of goals. We therefore changed in the Discussion “concrete goals” into “more concrete goals” and added in the Strengths and limitations section information about the difference between concrete individual goals and an overview.

Reviewer: 2

Reviewer Name: Christiane Muth

Institution and Country: Institute of General Practice, Johann Wolfgang Goethe University, Frankfurt / Main, Germany Please state any competing interests or state ‘None declared’: None declared.

Please leave your comments for the authors below The manuscript addresses a relevant topic of increasing awareness, as goal orientation is a necessary ingredient to patient-centred care. It is clearly and concise written, although I had some doubts (e.g., whether "freedom" was the correct translation of a verbatim or whether "autonomy" would suit better; as I am a non-native, I'd prefer to leave this open to a native). Also, the formal reporting quality was fine and in accordance with recommended standards.

Response: We discussed the suggestion of the reviewer of using “autonomy” instead of “freedom” with the native speaker who helped us with editing our text. According to him “freedom” was the best translation in this context, we therefore maintained “freedom”. We do, however, think “autonomy” is a

good suggestion for the category name. We therefore changed the category name “regaining/maintaining independence/freedom” into “regaining/maintaining autonomy”.

Major amendments:

1. Introduction section: The authors claim that their investigation was the first in hospital setting. Formally, this may be but what did the authors expect to find different/special in their setting in comparison to other settings (e.g., primary care and long-term care facilities)? In other words, what is already known about older patients' goals from other settings and what was the expected additional value to investigate patients' goals in hospital care? The background section could be improved by a more profound search for and summary of existing evidence (e.g., Vermunt et al. 2017, PMID: 28760149) to derive the rationale of the authors' objective.

Response: We are familiar with the study of Vermunt who aimed to identify the outcomes of shared decision making, however, this study does not give an overview of patient defined goals. In the Background we provided an overview of the existing evidence of goals of hospitalised patients, which is our population of interest. We are not aware of other studies giving an overview of goals of older patients.

2. Methods: Closely related to the issue of the setting is the approach to individual study participants. How did the authors recruit patients and what was the context when they asked them for participation (e.g., a day after admission, the day before discharge or at variable situations)? This may also explain why a patient gave a written informed consent but could not be interviewed due to a delirium. Did the authors intend to include patients with planned and unplanned admission and if so, why?

Response: Patients were recruited during their hospitalisation at variable situations, since it was not always possible in the beginning of the admission. We added information regarding the interview day in Table 1.

The patient with delirium was interviewed, because he signed informed consent, understood the purpose of my visit, was very interested to participate and recognized me and remembered very well the purpose of my visit the day we had an appointment for the interview, which was a day after I introduced myself and the study. According to the nurse he suffered from delirium when he was admitted, but the delirium was resolved according to her at time of the interview. However, during the interview appeared he still was sometimes confused and perhaps therefore not able to make his goals more specific than “getting better”.

We intended to include both planned and unplanned admissions because of the maximum variation sampling strategy. We added this information in the text about sampling.

3. Methods: The authors aimed at a saturated sample - what was the criterion for saturation? If the criterion was about patients' goals, was it realistic to achieve it (and a suitable sampling frame, therefore)?

Response: Our criterion for saturation was that no new information emerged from the patients. As we stated in the Discussion, this was not possible on goal level because there will always be new individual goals. We therefore decided to aim for saturation on category level, which appeared to be realistic. To clarify the criterion for saturation, we added some information in the Methods section.

4. Methods / introduction: what was the authors' definition of goals? Did they test / pilot their key question to assure that patients understand it?

Response: We added our definition in the text: "Goals are the personal health and life outcomes that people hope to achieve through their health care".

We did not pilot test our key question, since it was an open interview and we could clarify and probe. We only discovered that starting with the question "what do you hope to accomplish with this hospitalisation?", did not always work, some people felt caught off guard by that opening question. Therefore we started with giving the patient the opportunity to explain in his words the reason for hospitalisation, as we described in our methods section.

5. Methods: How was frailty assessed in the sample?

We added this in the Population paragraph: Frailty was determined by the Fried-criteria as operationalized by Avila Funes et al. (2008) and asked to the patient himself.

6. Methods / results / discussion: what do the authors mean with "probing"? What did they do to achieve answers about (health-related) goals from patients when interviews took up to 60 minutes? As this is also an important result, why didn't they present details from this "probing"-approach? Presented results could be further discussed - did the authors face similar difficulties like

Boeckxstaens et al. 2016 (PMID: 29090183) and if so, what is the impact for clinical implementation in goal setting?

Response: We agree with the reviewer and added in the methods section some examples of "probes".

As the reviewer suggested, we added an extra quote as an example of the probing process in the beginning of the results section:

Interviewer: Because what is your goal with this hospitalisation? Patient: Goal? Interviewer: Yes. Patient: That I am getting better. Interviewer: And what is better for you, can you describe that? Patient: Yes, that I ... well ... get my appetite back and drink well, because I am not interested in whether I get anything or not at the moment. I am not hungry, I am not thirsty and that has to change. Interviewer: Yes. Patient: And if I then grow stronger again. I have lost a lot of weight. From 88 to 82, I believe. Interviewer: In how much time? Patient: About a week. I was still very weak yesterday. Interviewer: Yes Yes. So grow stronger. Patient: To grow stronger. And that I am back on my feet,

that I can walk with a crutch and I'm done here as soon as possible and that I can go back home. That is my goal.

In the longer interviews the participant needed more time to explain the reason for admission and for example his living situation or daily functioning.

Regarding the article of Boeckxstaens et al. 2016 (PMID: 29090183): we want to thank the reviewer for sharing this interesting article with us. We recognise their findings partly. As we have written in our Results and Discussion, many patients were not familiar with goal-setting, but after probing, most were able to mention more concrete goals, so that last finding is incongruent with the findings of Boeckxstaens et al. 2016. Probably it was because we asked people in a concrete real life situation, namely a hospitalisation, where patients experience acute (worsening) of health problems

7. Results: Why did the authors decide to present this large number of categories, although there are some overlaps which might encourage other researchers to further collapse them (e.g. categories on controlling disease and improving conditions, on daily functioning and resuming hobbies / work, where codings list different types of (instrumental) activities of daily life)?

Response: In qualitative descriptive research, it is difficult to decide on which abstraction level to describe. We have chosen qualitative descriptive research because it stays closest to the participants (Neergaard, Olesen, Andersen, & Sondergaard, 2009; Sandelowski, 2000). However, if you stay too close to the participant (e.g. play cards with the neighbours), it is impossible to generalise. We therefore sought for an optimal point between generalisation and not too many codes and categories.

Regarding the examples of the reviewer: we think there is an important difference between controlling disease and improving condition, namely controlling disease is an objective medical outcome, while improving condition is a subjective experience by the patient. We added some extra information to the text to explain this.

Regarding daily functioning and resuming hobbies/ work: we agree with the reviewer that both are important activities for the patient, but the first considers only (I)ADL-activities and the second on a more participative level, which we think is important to distinguish.

8. Results: The presentation of the verbatims may support the results on 'comparing groups' better when more characteristics would be provided than just an ID and the age (e.g. whether this patient had a planned or unplanned admission). Further, I suppose that interviews were taken in Dutch but verbatims are presented in English - at what stage of analysis / reporting the verbatims were translated and by whom?

Response: Although we left out the section comparing groups' at the advice of the other reviewer, we think providing more patient information is a valuable addition. We therefore added information about acute/planned admission, internal medicine/surgery/cardiology and frailty.

Interviews and analysis were all in Dutch. The categories, codes and quotes were translated into English in the final stage by the first author and checked and edited by a native English speaker. To clarify this, we added this information in the Methods section.

9. Discussion - strengths and limitations: What may be the impact of a potential selection bias due to the recruitment / sampling process (> 90% of patients were included from the university hospital but <10% from the other hospital) on the results?

Response: We used a purposeful sampling strategy. Characteristic for purposeful sampling is that is not sought for statistically representativeness, but informational representativeness (Sandelowski, 1995). We discovered that not the type of hospital informed the patient goals, but the problems and priorities of the patient. We therefore think the fact that only <10% of the patients is recruited in a regional hospital did not affect the results.

Additional comment:

I am not familiar with rules and regulations in the Netherlands and was surprised that an ethics vote was not required. Therefore, I am not sure, whether any further data protection should be taken into consideration (to use pseudonyms not only for patients but also for the hospital names).

Response: We did present our study to the Medical Ethics Research Committee of the UMCG. This committee confirmed that the Medical Research Involving Human Subjects Act (WMO) did not apply to the research project. Research is subject to the WMO if the following criteria are met: 1. It concerns medical scientific research and 2. Participants are subject to procedures or are required to follow rules of behaviour (<https://english.ccmo.nl/>). Therefore the Medical Ethics Research Committee declared it did not have to give formal approval, but general privacy rules and agreements were still applicable, such as protection of identity, informed consent, etc.

Regarding the hospital names: we removed them from the text.

References

- Avila Funes, J. A., Helmer, C., Amieva, H., Barberger Gateau, P., Le Goff, M., Ritchie, K., . . . Dartigues, J. (2008). Frailty among community-dwelling elderly people in france: The three-city study. *The Journals of Gerontology. Series A, Biological Sciences and Medical Sciences*, 63(10), 1089-1096.
- Neergaard, M., Olesen, F., Andersen, R., & Sondergaard, J. (2009). Qualitative description - the poor cousin of health research? *BMC Medical Research Methodology*, 9, 52-52. doi:10.1186/1471-2288-9-52
- Patton, M. Q. (2002). *Qualitative research & evaluation methods* (3rd ed.). London: Sage Publications.
- Sandelowski, M. (1995). Sample size in qualitative research. *Research in Nursing & Health*, 18(2), 179-183.

Sandelowski, M. (2000). Whatever happened to qualitative description? Research in Nursing & Health, 23(4), 334-40. doi:10.1002/1098-240X(200008)23:43.0.CO;2-G

VERSION 2 – REVIEW

REVIEWER	Christiane Muth Institute of General Practice, Johann Wolfgang Goethe University, Frankfurt / Main, Germany
REVIEW RETURNED	03-May-2019

GENERAL COMMENTS	I am happy with the changes made. My concerns have been adequately addressed. The authors may consider to slightly change the header of the new Table 3: as it looks as if they present full preference patterns per patient (plus an example quotation), they might announce it to the readers. Also, these patterns add a nice goody to the paper.
---

VERSION 2 – AUTHOR RESPONSE

Reviewer(s)' Comments to Author:

Reviewer: 2

Reviewer Name: Christiane Muth

Institution and Country: Institute of General Practice, Johann Wolfgang Goethe University, Frankfurt / Main, Germany Please state any competing interests or state 'None declared': None declared.

Please leave your comments for the authors below I am happy with the changes made. My concerns have been adequately addressed.

The authors may consider to slightly change the header of the new Table 3: as it looks as if they present full preference patterns per patient (plus an example quotation), they might announce it to the readers. Also, these patterns add a nice goody to the paper.

Response: We changed the Table header into: Preference patterns and examples per participant . In addition, we announced the table in the text.